# New Azaphilones from the Marine-Derived Fungus *Penicillium sclerotiorum* E23Y-1A with Their Anti-Inflammatory and Antitumor Activities

**DOI:** 10.3390/md21020075

**Published:** 2023-01-22

**Authors:** Yanbo Zeng, Zhi Wang, Wenjun Chang, Weibo Zhao, Hao Wang, Huiqin Chen, Haofu Dai, Fang Lv

**Affiliations:** 1Hainan Provincial Key Laboratory for Functional Components Research and Utilization of Marine Bio-resources, Institute of Tropical Bioscience and Biotechnology, Chinese Academy of Tropical Agricultural Sciences & Key Laboratory for Biology and Genetic Resources of Tropical Crops of Hainan Province, Hainan Institute for Tropical Agricultural Resources, Haikou 571101, China; 2Beijing Key Laboratory for Separation and Analysis in Biomedicine and Pharmaceuticals, School of Life Science, Beijing Institute of Technology, Beijing 100081, China; 3Zhanjiang Experimental Station of Chinese Academy of Tropical Agricultural Sciences, Zhanjiang 524013, China

**Keywords:** marine-derived fungus, *Penicillium sclerotiorum*, azaphilones, structure elucidation, anti-inflammatory activity, antitumor activity

## Abstract

Nine new azaphilones, including penicilazaphilones I–N (**1**, **2** and **6**–**9**), *epi*-geumsanol D (**3**) and penidioxolanes C (**4**) and D (**5**) were isolated from the culture of the marine-derived fungus *Penicillium sclerotiorum* E23Y-1A. The structures of the isolates were deduced from extensive spectroscopic data (1D and 2D NMR), high-resolution electrospray ionization mass spectrometry (HRESIMS), and electronic circular dichroism (ECD) calculations. All the azaphilones from *P. sclerotiorum* E23Y-1A were tested for their anti-inflammatory and antitumor activities. Penicilazaphilone N (**9**) showed moderate anti-inflammatory activity with an IC_50_ value of 22.63 ± 2.95 μM, whereas penidioxolane C (**4**) exhibited moderate inhibition against human myeloid leukemia cells (K562), human liver cancer cells (BEL-7402), human gastric cancer cells (SGC-7901), human non-small cell lung cancer cells (A549), and human hela cervical cancer cells, with IC_50_ values of 23.94 ± 0.11, 60.66 ± 0.13, 46.17 ± 0.17, 60.16 ± 0.26, and 59.30 ± 0.60 μM, respectively.

## 1. Introduction

Marine-derived fungi represent an important source of marine natural products (MNPs). More than 33% of the new MNPs (approximately 2500) were reported from marine-derived fungi between 2016 and 2020 [1]. The interest in highly diverse structural classes and the number of various bioactivities of their metabolites keeps growing considerably [2,3,4,5]. Especially in some cases, marine fungi form symbiotic relationships with other organisms through the horizontal gene transfer of a mitochondrial intron from a fungus to marine organisms, consequentially fungi-derived natural products represent the characteristics of metabolites in symbiotic organisms as well [6,7,8].

*Penicillium* species, an important part of marine-derived fungi mainly derived from sediments, sponges, mangrove and algae, have a complex genetic background and abundant secondary metabolites. To date, more than 580 new natural products have been identified from marine-derived *Penicillium* fungi, including azaphilones, polyketides, alkaloids, terpenoids, and macrolides, which have displayed anti-inflammatory, antibacterial, anticancer, and other activities with potential application in the pharmaceutical and medical fields [9,10,11,12]. Azaphilones, a class of structurally diverse fungal metabolites mainly from *Penicillium* fungi, and defined as polyketides possessing a highly oxygenated pyranoquinone bicyclic core and a quaternary carbon center, have been found to exhibit various biological properties, such as antibacterial, antifungal, cytotoxic, anti-inflammatory, and enzyme inhibitory activities [13]. In our continuing interest in finding new compounds with potential bioactivities [14,15], a chemical investigation of the fungus *Penicillium sclerotiorum* E23Y-1A derived from the marine sponge *Holoxea* sp. was performed to yield a further nine new azaphilones, including penicilazaphilones I–N (**1**, **2** and **6**–**9**), *epi*-geumsanol D (**3**), and penidioxolanes C (**4**) and D (**5**) (Figure 1). The structures of the new compounds were determined by comprehensive analyses of NMR spectra, HRESIMS data, and ECD calculation. By screening of the inhibitory effects on NO production in the LPS-induced RAW 264.7 macrophages, the anti-inflammatory activity of these compounds was evaluated, and the results showed that compound **9** exhibited moderate inhibitory activity with an IC_50_ value of 22.63 ± 2.95 μM. Cytotoxic activities showed that penidioxolane C (**4**) exhibited moderate inhibition against K562, BEL-7402, SGC-7901, A549 and Hela cancer cells with IC_50_ values of 23.94 ± 0.11, 60.66 ± 0.13, 46.17 ± 0.17, 60.16 ± 0.26, and 59.30 ± 0.60 μM, respectively. This article describes the isolation, structure elucidation, as well as anti-inflammatory activity and cytotoxicities of these new azaphilones.

## 2. Results and Discussion

### 2.1. Structure Elucidation of New Compounds

Penicilazaphilone I (**1**) was obtained as a yellow oil. The molecular formula of **1** was established as C_22_H_32_O_7_ with seven degrees of unsaturation according to HRESIMS data at *m/z* 431.2038 (calcd. 431.2040 for C_22_H_32_O_7_Na, [M + Na]^+^), which was confirmed by the ^13^C NMR and DEPT spectral data. The ^13^C NMR and DEPT data of compound **1** (Table 1 and Table 2) suggested the presence of 22 carbons, comprising six methyl carbons (*δ*_C_ 50.4, 20.8, 20.0, 16.4, 14.0 and 11.9), two methylene carbons (*δ*_C_ 68.0 and 29.0), eight methine carbons (*δ*_C_ 139.5, 125.0, 116.9, 105.0, 79.5, 74.5, 35.4 and 34.7) and six quaternary carbons (*δ*_C_ 194.6, 170.6, 159.6, 151.1, 80.4 and 74.4). The HMBC correlations from H-1 (*δ*_H_ 4.35, 3.80) to C-3 (*δ*_C_ 159.6) and C-4a (*δ*_C_ 151.1), from H-4 (*δ*_H_ 5.67) to C-5 (*δ*_C_ 116.9) and C-8a (*δ*_C_ 34.7), from H-5 (*δ*_H_ 5.80) to C-7 (*δ*_C_ 74.4), from H-8 (*δ*_H_ 4.98) to C-4a, the ketocarbonyl carbon C-6 (*δ*_C_ 194.6) and C-7, and from H-8a (*δ*_H_ 3.35) to C-4a, established a bicyclic core moiety [16]. Further analysis of HMBC correlations from H_3_-16 (*δ*_H_ 1.36) to C-6 and the oxygenated carbon C-8 (*δ*_C_ 74.5), from H_3_-20 (*δ*_C_ 2.21) to the ketocarbonyl carbon C-19 (*δ*_C_ 170.6), and from H-8 to C-19, indicated a methyl group and an acetoxy group attached at C-7 and C-8, respectively. These analysis of NMR spectra confirmed the characteristic signals of an azaphilone skeleton for compound **1**. The ^1^H-^1^H COSY spectrum of compound **1** showed two separated correlations of H-9/H-10 and H-12/H-13/H-14 (H_3_-18)/H_3_-15 in the side chain as well as one correlation of H-8/H-8a/H-1 in the bicyclic core (Figure 2). The ^1^H-^1^H COSY correlations of H-9/H-10 and H-12/H-13/H-14 (H_3_-18)/H_3_-15, associated with HMBC correlations from H-10 (*δ*_H_ 6.31) to C-9 (*δ*_C_ 125.0) and C-11 (*δ*_C_ 80.4), from H-12 to C-10 (*δ*_C_ 139.5) and C-14 (*δ*_C_ 29.0), from H_3_-18 (*δ*_H_ 0.90) to C-12 (*δ*_C_ 79.5) and C-14, together with the correlations from H_3_-OCH_3_ (*δ*_H_ 3.16) to C-11, H_3_-17 (*δ*_H_ 1.29) to C-10, C-11 and C-12, established a 3-methoxy-3,5-dimethylhept-1-ene-4-ol unit. In addition, HMBC correlations from H-9 to C-3 and C-4 showed that the whole side chain was attached at C-3 of the azaphilone unit. All arrangements of 1D and 2D NMR data of compound **1** led to the construction of the planar structure of compound **1**.

The relative configuration of 7*R**, 8*R**, 8a*R**of compound **1** was suggested by NOESY correlations and ^1^H-^1^H coupling constants [17]. A large coupling constant (^3^*J_H-_*_8/ *H-*8a_ = 10.0 Hz) suggested that H-8 and H-8a were positioned on the opposite face [17]. Meanwhile, the NOESY correlation between H_3_-16 and H-8 indicated their cofacial orientation (Figure 3). As to azaphilones possessing a branched C_7_ side chain fused at C-3, the absolute configuration of C-13 was suggested as *S* by the common biosynthetic pathway of the aliphatic side chain [13,18,19], which was unambiguously defined by X-ray single-crystal diffraction [20], hydrolysis [21], or ECD calculation [19]. To determine the relative configuration of C-11 and C-12 in the side chain of **1**, we performed theoretical NMR chemical shifts calculations of four diastereomers (Figure 4) of **1** at mPW1PW91-SCRF/6-311G(d,p)//B3LYP-D3BJ/6-31G(d) theoretical level in methanol with the GIAO method [22]. The calculated ^13^C and ^1^H NMR chemical shifts of (11*R**, 12*R**, 13*S**)-**1** showed the best agreement with the experimental values. Furthermore, DP4+ analysis [23] predicted that (11*R**, 12*R**, 13*S**)-**1** was the most likely candidate with 91.85% probability (Appendix A, Appendix A). Considering these observations, the relative stereochemistry of **1** was defined as depicted in Figure 1. The absolute configuration of (7*R*, 8*R*, 8a*R*, 11*R*, 12*R*, 13*S*) of compound **1** was established by analyzing the CD curve (Figure 5) and biosynthetic considerations. Cotton effects at 232 nm, 261 nm and 377 nm were completely consistent with those of penicilazaphilone F [14] because the substitution at C-5 in azaphilone had no effect on the configuration [24]. To further verify the assigned absolute configuration of compound **1**, its theoretical ECD curve was calculated. The calculated ECD curve was consistent with the experimental one (Figure 6), confirming the stereochemical assignment for compound **1**.

In order to conveniently clarify the configurations of co-metabolites (compounds **2–9**) of compound **1**, the molecular orbital (MO) analysis of the most populated conformer of compound **1** was carried out to reveal that the positive Cotton effect around 266 nm was assigned to *n*→*π** electron transition from acetate carbonyl to conjugated azaphilone unit, dominated by MO 105→MO 111 transition, and the negative Cotton effect around 375 nm was related to π→π* electron transition of conjugated azaphilone unit from MO 110 (HOMO)→MO 111 (LUMO) transition (Appendix A). These two characteristic Cotton effects did not involve the electron transition of the functional groups on the side chain. Therefore, the ECD curves actually established the configurations of stereocenters on the azaphilone moiety, while the configurations of stereocenters on the side chain of co-metabolites were established by their NOESY correlation, ^1^H-^1^H coupling constants, chemical shifts and Cotton effects with those of the known analogues, in combination with biosynthetic considerations.

Penicilazaphilone J (**2**) was obtained as a yellow oil and assigned the same molecular formula of C_22_H_32_O_7_ as that of compound **1** by HRESIMS data at *m/z* 839.4156 (calcd. for 839.4188, C_44_H_64_O_14_Na, [2M + Na]^+^) and the ^13^C NMR and DEPT spectral data. Careful comparation of their 1D and 2D NMR spectra revealed that compound **2** had the same planar structure as that of compound **1**. However, the obvious difference at C-17 (△*δ*_C_ = 2.1 ppm) was observed (Table 2) and the NOESY spectrum of compound **2** exhibited a NOE correlation between H_3_-OCH_3_ and H-12 instead of the correlation of H_3_-17 and H-12 of compound **1**, which suggested H_3_-OCH_3_ and H-12 of compound **2** were located on the same side. However, relative configuration assignment only based on NOESY correlations is usually not reliable enough in conformationally flexible molecules such as the acyclic part of organic compounds. To irrefutably determine the relative configuration of C-11 and C-12 on the side chain of **2**, the quantum chemistry–nuclear magnetic resonance calculations were employed. Following the DP4+ protocol, the calculated results showed that the experimentally observed NMR data for **2** gave a better match of the 11*S**, 12*R**, 13*S** isomer with 99.67% probability, while the isomer 11*R**, 12*R**, 13*S** resulted at 0.3% probability (Appendix A, Appendix A). Consequently, the relative configuration of the chain side of compound **2** was established as 11*S**, 12*R**, 13*S**. Among the hitherto reported azaphilones possessing a branched C_7_ side chain fused at C-3, the absolute configuration of C-13 in the aliphatic side chain was *S*, which suggests the aliphatic branch of the azaphilones originates from a shared biosynthetic pathway [13]. Thus, the absolute configuration of C-13 of compound **2** was assigned the *S* configuration. Consequently, the absolute configuration of the chain side of compound **2** was established as 11*S*, 12*R*, 13*S*. As compounds **1** and **2** were co-metabolites showing similar Cotton effects (Figure 5), the absolute stereochemistry of the azaphilone core moiety in compound **2** was proposed to be 7*R*, 8*R*, and 8a*R*, the same as that of compound **1**. Therefore, compound **2** was identified as a chiral isomer of compound **1** at C-11.

*Epi*-geumsanol D (**3**) was obtained as a yellow oil, its molecular formula C_21_H_30_O_7_ was assigned by analysis of HRESIMS at *m/z* 417.1863 (calcd. for 417.1884, C_21_H_30_O_7_Na, [M + Na]^+^) in combination with NMR data. Careful comparison of the ^1^H and ^13^C NMR data of compounds **3** and **2**, found that compound **3** was highly similar to compound **2** except for the absence of a methoxyl group at C-11 (Table 1 and Table 2). A large coupling constant (^3^*J_H-_*_8/*H-*8a_ = 10.0 Hz) suggested that H-8 and H-8a were positioned on the opposite face [17]. Meanwhile, the NOESY correlation between H_3_-16 and H-8 indicated their cofacial orientation. The ^3^*J*_H-12/H-13_ Value (2.1 Hz) was indicative of a *gauche* relationship of H-12 and H-13. The NOESY correlation between H-12 and H-13, in association with the absence of an NOESY correlation between H_3_-17 and H-12 suggested the relative configuration at the side chain of compound **3** to be 11*S**, 12*R**, and 13*S** [25,26,27]. In order to irrefutably confirm the relative configuration at the side chain of compound **3**, the NMR calculation was performed and followed by a DP4+ analysis. The calculated chemical shifts of (11*S**, 12*R**, 13*S**)-**3** showed best agreement with the experimental values among the possible diastereomers and (11*S**, 12*R**, 13*S**)-**3** possessed 100% DP4+ probability (Appendix A, Appendix A), indicating that the relative configuration at the side chain of compound **3** to be 11*S**, 12*R**, and 13*S**. Furthermore, similar Cotton effects at 245 nm, 280 nm and 377 nm to compound **2** (Figure 5), in combination with biosynthetic considerations, allowed the absolute configuration of compound **3** to be 7*R*, 8*R*, 8a*R*, 11*S*, 12*R*, and 13*S*.

**Table 1 marinedrugs-21-00075-t001:** ^1^H NMR data of compounds **1**–**9** in CDCl_3_.

No.	1 *^a^*	2 ^*a*^	3 *^b^*	4 ^*a*^	5 *^a^*	6 ^*a*^	7 *^a^*	8 ^*a*^	9 ^*a*^
1	4.35, dd (10.8, 5.2)	4.35, dd (10.8, 5.2)	4.35, dd (10.7, 5.2)	4.36, dd (10.7, 5.3)	4.35, dd (10.8, 5.2)	4.86, dd, (10.9, 5.5)	4.39, dd, (10.8, 5.3)	4.28, dd, (10.7, 5.3)	7.41, s
	3.80, dd (13.7,10.8)	3.80, dd (13.6, 10.8)	3.81, dd (13.6,10.7)	3.79, dd (13.6, 10.7)	3.80, dd (13.6, 10.7)	3.81, dd, (13.4, 10.9)	3.82, dd, (13.6, 10.9)	3.73, dd, (13.8, 10.8)	
4	5.67, s	5.67, s	5.67, s	5.67, s	5.68, s	5.94, s	5.96, s	5.52, s	6.37, s
5	5.80, brs	5.79, s	5.79, d (2.1)	5.80, d (2.1)	5.79, d (2.1)	5.88, brs	5.91, brs	5.70, s	5.53, s
8	4.98, d (10.0)	4.97, d (10.1)	4.98, d (10.0)	4.98, d (10.1)	4.98, d (10.0)	3.47, d (9.3)	4.99, d (10.0)	4.95, d (9.9)	3.85, d (12.2)
8a	3.35, m	3.35, m	3.35, m	3.34, m	3.34, m	3.04, m	3.37, m	3.27, m	
9	6.00, d (15.9)	5.98, d (16.1)	6.19, d (15.6)	6.12, d (15.5)	6.18, d (15.4)	6.85, d (15.6)	6.84, d (15.7)	2.51, t (7.3)	6.93, d (15.8)
10	6.31, d (15.9)	6.42, d (16.1)	6.43, d (15.5)	6.37, d (15.5)	6.39, d (15.4)	6.70, d (15.7)	6.65, d (15.7)	2.68, m	6.76, d (15.7)
12	3.39, d (2.7)	3.45, s	3.49, d (2.1)	3.38, d (8.8)	3.30, d (9.6)	2.33, s	1.36, s	2.17, s	2.36, s
13	1.53, m	1.47, m	1.53, m	1.47, m	1.64, m	1.49, s	1.49, s	1.34, s	1.58, s
14	1.38, m	1.40, m	1.41, m	1.62, m	1.39, m				
	1.24, m	1.28, m	1.30, m	1.07, m	1.01, m				
15	0.85, t (7.4)	0.87, t (7.4)	0.90, t (7.4)	0.88, t (7.4)	0.90, t (7.3)		2.23, s	2.20, s	3.75, d (12.3)
16	1.35, s	1.34, s	1.35, s	1.36, s	1.36, s				
17	1.29, s	1.31, s	1.35, s	1.41, s	1.24, s				2.48, s
18	0.90, d (6.8)	0.83, d (6.5)	0.86, d (6.8)	0.97, d (6.6)	1.04, d (6.5)				
20	2.21, s	2.20, s	2.21, s	2.21, s	2.21, s				
OCH_3_	3.16, s	3.16, s							
21				4.71, d (5.0)	4.74, d (4.0)				
22				1.62, m	1.60, m				
23				1.60, m	1.60, m				
				1.20, m	1.20, m				
24				0.93, t (7.3)	0.93, t (7.3)				
25				0.98, d (6.6)	0.94, d (6.6)				

*^a^* in 600 MHz; *^b^* in 400 MHz.

Penidioxolane C (**4**) was obtained as a yellow oil. The molecular formula, C_26_H_38_O_7_ with eight degrees of unsaturation, was established by HRESIMS at *m/z* 463.2683 (calcd. for 463.2690, C_26_H_39_O_7_, [M + H]^+^) in combination with its NMR data. The ^13^C NMR and DEPT data of compound **4** (Table 1 and Table 2) suggested the presence of 26 carbons, comprising seven methyl carbons (*δ*_C_ 25.2, 20.8, 20.1, 16.0, 14.1, 11.6 and 11.2), three methylene carbons (*δ*_C_ 68.1, 25.6 and 24.8), ten methine carbons (*δ*_C_ 138.9, 123.2, 116.7, 105.3, 104.8, 90.4, 74.5, 38.8, 35.2 and 34.8) and six quaternary carbons (*δ*_C_ 194.4, 170.6, 160.0, 151.2, 80.8 and 74.5). Investigation of ^1^H-^1^H COSY correlations of H-8/H-8a/H-1 and HMBC correlations from H-1 to C-3/ C-4a, from H-4 to C-5/C-8a, from H-5 to C-7, from H-8 to C-6, from H-8a to C-4a, together with HMBC correlations from H_3_-16 to C-6/C-8, from H_3_-20 to C-19, and from H-8 to C-19 indicated that compound **4** possessed the same azaphilone skeleton as that of compound **1**. However, the side chain of compound **4** included the additional consecutive ^1^H-^1^H COSY correlations of H-21/H-22/H_2_-23 (H_3_-25)/H_3_-24 excepting H-9/H-10 and H-12/H-13/H-14 (H_3_-18)/H_3_-15 were identical to compound **1**. This observation was further confirmed by the HMBC correlations from H-24 to C-23, from H-23 to C-22, from H-21 to C-22, and from H-25 to C-21, no long range HMBC correlations from H-21 to C-11 and C-12 were found, and the connection position with the azaphilone unit was deduced by downfield chemical shifts of C-11 and C-12 [28] (Figure 2). All arrangements of 1D and 2D NMR data allowed the construction of the planar structure of compound **4**. The relative configuration of compound **4** was suggested by ^1^H-^1^H coupling constants [16], together with NOESY spectral analyses. A large coupling constant (^3^*J*_8/8a_ = 10.1 Hz) and NOESY correlation between H_3_-16 and H-8 indicated H-8 should be placed on the opposite orientation of H-8a with a cofacial orientation of H_3_-16 [17]. The C-7, C-8 and C-8a chirality centers in the bicyclic core were assigned as 7*R**, 8*R**, 8a*R**. The relative configuration of 1,3-dioxolane part of compound **4** was determined as 11*S**, 12*R*,* 13*S**, 21*R**, and 22*R** by NOESY correlations of H-12 (*δ*_H_ 3.38)/H_3_-17 (*δ*_H_ 1.41), H-12 (*δ*_H_ 3.38)/H_3_-18 (*δ*_H_ 0.97), H-12 (*δ*_H_ 3.38)/H-21 (*δ*_H_ 4.71), H-21 (*δ*_H_ 4.71)/H_3_-17 (*δ*_H_ 1.41), and H-21 (*δ*_H_ 4.71)/ H_3_-25 (*δ*_H_ 0.98) (Figure 3), which was irrefutably confirmed by NMR calculation followed by DP4+ analysis. The calculated chemical shifts of (11*S**, 12*R*,* 13*S**, 21*R**, 22*R**)-**4** showed best agreement with the experimental values among the possible diastereomers and (11*S**, 12*R*,* 13*S**, 21*R**, 22*R**)-**4** possessed 99.83% DP4+ probability (Appendix A, Appendix A). Thus, the relative stereochemistry of **4** was unambiguously defined as depicted in Figure 1. The absolute configuration of C-13 was suggested as *S* by the common biosynthetic pathway of the aliphatic side chain in azaphilones. Consequently, the absolute configuration of the 1,3-dioxolane part of compound **4** was assigned as 11*S*, 12*R,* 13*S*, 21*R*, and 22*R*. As compound **4** was a co-metabolite of compound **1** with similar Cotton effects (Figure 5), the stereochemistry of the azaphilone core moiety in compound **4** was proposed to be 7*R*, 8*R*, and 8a*R* as well. Therefore, the absolute configuration of compound **4** was 7*R*, 8*R*, 8a*R*, 11*S*, 12*R,* 13*S*, 21*R*, and 22*R*.

**Table 2 marinedrugs-21-00075-t002:** ^13^C NMR data (125 MHz) of compounds **1**–**9** in CDCl_3_.

No.	1	2	3	4	5	6	7	8	9
1	68.0	68.0	68.1	68.1	68.0	69.1	68.2	68.3	147.1
3	159.6	159.6	160.0	160.0	160.0	158.1	157.7	166.4	153.5
4	105.0	104.8	104.7	104.8	105.0	110.9	110.7	101.3	114.3
4a	151.1	151.1	151.1	151.2	151.0	149.8	149.2	150.9	142.8
5	116.9	116.9	116.7	116.7	116.7	119.1	119.5	115.5	109.1
6	194.6	194.6	194.4	194.4	194.4	196.7	194.5	194.5	191.8
7	74.4	74.4	74.4	74.5	74.5	74.5	74.6	74.4	82.8
8	74.5	74.5	74.6	74.5	74.5	74.4	74.3	74.5	42.9
8a	34.7	34.7	36.4	34.8	34.8	36.7	34.7	34.4	114.4
9	125.0	125.0	122.3	123.2	122.5	135.0	134.8	28.5	131.9
10	139.5	138.4	140.0	138.9	140.8	130.3	130.4	40.1	131.0
11	80.4	80.6	76.1	80.8	80.7	197.8	197.7	206.6	196.8
12	79.5	80.2	79.4	90.4	88.8	28.7	28.5	30.1	29.0
13	35.4	35.7	34.8	35.2	34.5	20.5	19.9	20.1	23.3
14	29.0	29.2	28.9	25.6	25.4		170.5	170.6	168.3
15	11.9	12.0	12.0	11.2	11.1		20.8	20.8	57.2
16	20.0	20.0	20.1	20.1	20.1				200.0
17	16.4	18.5	26.7	25.2	21.3				30.3
18	14.0	13.8	13.4	16.0	16.4				
19	170.6	170.6	170.6	170.6	170.5				
20	20.8	20.8	20.8	20.8	20.8				
OCH_3_	50.4	50.6							
21				105.3	105.8				
22				38.8	38.7				
23				24.8	24.5				
24				11.6	11.8				
25				14.1	13.6				

The molecular formula of penidioxolane D (**5**) was identical to that of compound **4** as determined by HRESIMS at *m/z*: 463.2688 (calcd. for 463.2690, C_26_H_39_O_7_, [M + H]^+^). Detailed analysis of NMR spectra of compound **5** revealed that it shared the same planar structure as that of compound **4**. The major changes in the ^13^C NMR data of compound **5** were the shielded signal of C-12 (−1.6 ppm) and the de-shielded signals of C-10 (+1.9 ppm) as well as the shielded signal of H-17 (−0.17 ppm) in ^1^H NMR data when compared with compound **4**. Moreover, in the NOESY spectrum of compound **5**, combined with the correlation of H-12 (*δ*_H_ 3.30) and H-21 (*δ*_H_ 4.74), the lack of the correlations of H_3_-17 (*δ*_H_ 1.24) and H-12, and H_3_-17 and H-21, implied H_3_-17 was the opposite orientation of H-12 and H-21 in compound **5**. These observations suggested the configuration of C-11 of compound **5** was different from that of compound **4**. Consequently, the configuration of C-11 of compound **5** was confirmed as *R*.

Penicilazaphilone K (**6**) was obtained as a yellow oil. The molecular formula, C_14_H_16_O_5_, was established by the protonated molecular ion peak at *m/z* 265.1073 (calcd. for C_14_H_17_O_5_, 265.1071, [M + H]^+^) in the HRESIMS, indicating seven degrees of unsaturation. Analysis of ^1^H NMR spectrum (Table 1) of compound **6** revealed the presence of two methyls (*δ*_H_ 2.33, 1.49), one methylene (*δ*_H_ 4.86, 3.81), two methines (*δ*_H_ 3.47, 3.04), and four olefinic protons (*δ*_H_ 6.85, 6.70, 5.94, 5.88). The ^13^C NMR and DEPT spectra of compound **6** revealed 14 carbon resonances, including two methyls (*δ*_C_ 28.7, 20.5), one methylene (*δ*_C_ 69.1), two methines (*δ*_C_ 74.4, 36.7), six olefinic carbons (*δ*_C_ 158.1, 149.8, 135.0, 130.3, 119.1, 110.9), and three quaternary carbons (*δ*_C_ 197.8, 196.7, 74.5) (Table 2). In addition to a 1-butene-3-one unit (four carbons), the remaining 10 carbons could be assigned to a typical azaphilone skeleton with a methyl group at C-7 confirmed by HMBC correlations from H-1 to C-3 and C-4a, from H-4 to C-5 and C-8a, from H-8 to C-8a and C-1, and from H_3_-13 to C-6 and C-8. The key ^1^H-^1^H COSY and HMBC correlations of compound **6** are shown in Figure 2. The relative configuration of 7*R**, 8*R**, 8a*R** was established by NOESY correlation and ^1^H-^1^H coupling constant. A large coupling constant (^3^*J*_8/8a_ = 9.3 Hz) suggested that H-8 and H-8a are positioned on the opposite face [17]. The NOESY correlation between H_3_-13 and H-8, meanwhile, indicated their cofacial orientation (Figure 3). Furthermore, the absolute configuration of compound **6** was established by its CD spectrum (Figure 5). The calculated ECD curve was consistent with the experimental one (Figure 6), confirming the stereochemical assignment of (7*R*, 8*R*, 8a*R*) for compound **6**.

Penicilazaphilone L (**7**) was obtained as a yellow oil. The molecular formula C_16_H_18_O_6_ with eight degrees of unsaturation was established by analysis of HRESIMS at *m/z* 307.1180 (calcd. for 307.1176, C_16_H_19_O_6_, [M + H]^+^) in combination with its NMR data. The NMR data of compound **7** were quite similar to those of compound **6**, with notable difference in the ^1^H, ^13^C, and HSQC NMR data, being the presence of an additional acetoxy group (*δ*_C_ 170.5, *δ*_C_ 20.8, *δ*_H_ 2.23) (Table 1 and Table 2). The assignment of an acetoxy group was performed by the HMBC correlation from H_3_-15 (*δ*_H_ 2.23) to the ester carbonyl C-14 (*δ*_C_ 170.5). The HMBC correlation from the oxymethine proton H-8 (*δ*_H_ 4.99) to C-14 indicated the acetoxy group connected with the oxymethine carbon C-8. The analysis of NOESY correlations and CD curve (Figure 5) determined that the configuration of compound **7** was 7*R*, 8*R*, 8a*R*, the same as that of compound **6**.

Penicilazaphilone M (**8**) was obtained as a yellow oil. The molecular formula C_16_H_20_O_6_ with seven degrees of unsaturation was established by the analysis of HRESIMS (*m/z* 331.1141 calcd. for 331.1152, C_16_H_20_NaO_6_, [M + Na]^+^) in combination with its NMR data. Analysis of the 1D and 2D NMR data of compound **8** allowed construction of a structure similar to that of compound **7** with the disappearance of one double bond (*∆*^9,10^). A large coupling constant (^3^*J*_8/8a_ = 9.9 Hz) and a NOESY correlation between H_3_-13 and H-8 allowed the determination of the relative configuration of compound **7** [17]. The absolute configuration of (7*R*, 8*R*, 8a*R*) of compound **8** was confirmed by its CD curve (Figure 5) showing the same characteristics as that of compound **1**.

Penicilazaphilone N (**9**) was obtained as a yellow amorphous solid. The molecular formula C_18_H_16_O_6_ with 11 degrees of unsaturation was established by analysis of HRESIMS (*m/z* 351.0835 [M + Na]^+^) and NMR data (Table 1 and Table 2). The azaphilone skeleton of compound **9** was established by comparisons of NMR data with those of compound **6**. In addition to the appearance of a double bond (*∆*^1,8a^), further analysis of ^13^C NMR and HSQC spectra with those of compound **6** determined the presence of four additional carbon resonances, including two carbonyls (*δ*_C_ 200.0 and 168.3), one methine (*δ*_C_ 57.2), and one methyl (*δ*_C_ 30.3). The ^1^H-^1^H COSY correlation between H-8 (*δ*_H_ 3.85) and H-15 (*δ*_H_ 3.75), along with HMBC correlations from H-8 to C-7 (*δ*_C_ 82.8) and from H-15 to C-14 (*δ*_C_ 168.3), confirmed the presence of the *γ*-lactone ring (Figure 2). Furthermore, HMBC correlations from H_3_-17 (*δ*_H_ 2.48) to C-15 (*δ*_C_ 57.2) and C-16 (*δ*_C_ 200.0) determined an acetyl group attached to C-15. The relative configuration of 7*R**, 8*R**, 15*R** was established by a large coupling constant (^3^*J*_8/15_ = 12.2 Hz) and NOESY correlation between H-8 and H_3_-13, which suggested that H-8 and H-15 are positioned on the opposite face while H-8 and H_3_-13 have a cofacial orientation (Figure 3). The absolute configurations of compound **9** could not be established by direct comparison with the CD curve of compound **9,** as for other compounds, due to the presence of the *γ*-lactone moiety. Therefore, the theoretical ECD curve of compound **9** was calculated (Figure 5). Its calculated ECD curve of (7*R*, 8*R*, and 15*R*) configuration was consistent with the experimental one (Figure 6), which confirmed the absolute configurations of **9** as shown.

All of the isolated compounds (**1**–**9**) possessed a common azaphilone framework possessing a highly oxygenated pyranoquinone bicyclic core and a quaternary carbon center and their co-occurrence in the same fungus suggested that they should originate from the same biogenetic pathway. A plausible biosynthetic pathway toward the formation of compounds **1**–**9** can be proposed by detailed analysis of their structures (Figure 1).

### 2.2. Bioassay

All compounds were tested for anti-inflammatory activity and antitumor activities in vitro. The anti-inflammatory activity test showed that compound **9** exhibited moderate inhibition of nitric oxide production in LPS-stimulated RAW264.7 cells with an IC_50_ value of 22.63 ± 2.95 μM, and compounds **1**, **6** and **7** exhibited weak inhibition with IC_50_ values of 50.71 ± 8.41, 65.30 ± 7.21 and 31.84 ± 2.79 μM, respectively (Figure 7). Quercetin as a positive control showed an IC_50_ value of 11.19 ± 0.38 μM. Further research on cell proliferation was carried out by MTT method, and the result indicated compounds **1**, **6**, **7** and **9** had no cytotoxicity against RAW264.7 at a concentration of 100 μM.

The isolated azaphilones were evaluated for their cytotoxicities against human myeloid leukemia cells (K562), human liver cancer cells (BEL-7402), human gastric cancer cells (SGC-7901), human non-small cell lung cancer cells (A549), and human hela cervical cancer cells. The results were presented in Table 3. Penidioxolane C (**4**) exhibited moderate inhibition against K562, BEL-7402, SGC-7901, A549, and Hela cancer cells with IC_50_ values of 23.94 ± 0.11, 60.66 ± 0.13, 46.17 ± 0.17, 60.16 ± 0.26, and 59.30 ± 0.60 μM, respectively. Cisplatin as a positive control showed IC_50_ values of 3.08 ± 0.05, 4.02 ± 0.06, 4.11 ± 0.02, 1.93 ± 0.02, and 11.29 ± 0.15 μM to the abovementioned cancer cells, respectively.

## 3. Materials and Methods

### 3.1. General Experimental Procedures

Optical rotation was performed on a Perkin–Elmer 241MC polarimeter (PerkinElmer, Fremont, CA, USA). NMR data were measured using Bruker Avance III 400 and Bruker DRX 600 instruments (Bruker Biospin AG, Fällanden, Germany). All chemical shifts (δ) were given in ppm referenced to TMS and coupling constants (J) given in Hz. HRESIMS were recorded on an Agilent G6520 Q-TOF mass spectrometer. The ECD curves and UV data were collected on a Jasco J-810 spectropolarimeter (JASCO, Tokyo, Japan). IR absorptions were obtained on a Nicolet 380 FT-IR instrument (Thermo, Waltham, MA, USA) using KBr pellets. HPLC was carried out using an Agilent 1260 series liquid chromatography (Agilent Technology Co., Ltd., Santa Clara, CA, USA) equipped with a DAD G1315D detector and an Agilent Eclipse XDB-C_18_ column (5 µm, 9.4 × 250 mm). Column chromatography was performed on a Sephadex LH-20 (Merck, Darmstadt, Germany) and silica gel (200–300 and 300–400 mesh; Qingdao Haiyang Chemical Group Co., Ltd., Qingdao, China).

### 3.2. Fungal Identification, Fermentation, and Extract

The fungus was isolated from the marine sponge Holoxea sp. collected in March 2019 from Quanfu Island, Hainan, China. It was authenticated as Penicillium sclerotiorum E23Y-1A (GenBank accession No. MW090660) via DNA amplification and sequencing of the internal transcribed spacer region of the rRNA gene, along with morphological characteristics. The fungus strain was deposited at the Hainan Provincial Key Laboratory for Functional Components Research and Utilization of Marine Bio-resources, Haikou, China.

The fungus *P*. *sclerotiorum* E23Y-1A was cultured in eighty-three 1000 mL Erlenmeyer flasks at room temperature for 28 days. Each flask contained 80 g rice and 120 mL water with 3.3% NaBr. After fermentation, the cultures were extracted four times with EtOAc, and 192.0 g dark brown gum was obtained.

### 3.3. Isolation and Purification

All extract was fractionated by vacuum liquid chromatography using silica gel as a stationary phase eluted with a step gradient of petroleum ether-EtOAc (1:0, 20:1, 10:1, 8:2, 7:3, 6:4 5:5, 3:7 and 0:1; *v*/*v*) and CH_2_Cl_2_-MeOH (9:1, 8:2 and 0:1; *v*/*v*) to obtain fractions 1–25. Fraction 14 (30.0 g) was separated by Sephadex LH-20 column chromatography eluted with MeOH and then purified by silica gel column chromatography eluted with CH_2_Cl_2_-MeOH (4:1; *v*/*v*) to give compound **9** (2.9 mg). Fraction 18 (5.3 g) was purified by Sephadex LH-20 column chromatography eluted with petroleum ether-CH_2_Cl_2_-MeOH (2:1:1, *v*/*v*/*v*) to afford subfractions N1-N14. Subfraction N5 (46.9 mg) was separated by silica gel column chromatography eluted with CH_2_Cl_2_-acetone (80:1; *v*/*v*) and further purified by semi-preparative HPLC using MeOH-H_2_O (80:20, *v*/*v)* as mobile phase to afford compounds **4** (2.9 mg, *t*_R_ = 19.0 min) and **5** (2.7 mg, *t*_R_ = 20.0 min). Subfraction N7 (327.0 mg) was separated by silica gel column chromatography eluted with CH_2_Cl_2_-acetone (70:1; *v*/*v*) and further purified by semi-preparative HPLC using MeOH-H_2_O (60:40, *v*/*v*) as mobile phase to afford compounds **1** (8.6 mg, *t*_R_ = 16.0 min) and **2** (12.6 mg, *t*_R_ = 17.0 min). Subfraction N10 (1.9 g) was separated by silica gel column chromatography eluted with CH_2_Cl_2_-acetone (70:1; *v*/*v*) and further purified by semi-preparative HPLC to afford compound **3** (5.5 mg, *t*_R_ = 13.8 min) by MeOH-H_2_O (55:45, *v*/*v*), compounds **7** (3.5 mg, *t*_R_ = 16.0 min) and **8** (2.2 mg, *t*_R_ = 18.0 min) by MeOH-H_2_O (42:58, *v*/*v*), and compound **6** (3.5 mg, *t*_R_ = 17.0 min) by MeOH-H_2_O (35:65, *v*/*v*).

Penicilazaphilone I (**1**): yellow oil; [*α*]^25^_D_ −193.1 (*c* 0.1, MeOH); UV (MeOH) *λ*_max_ (log*ε*) 245 (2.96), 350 (3.63) nm; IR (KBr) *ν*_max_ 3419, 2931, 1745, 1661, 1586, 1379, 1229, 1060 cm^−1^; HRESIMS *m/z*: 431.2038 [M + Na]^+^ (calcd. for 431.2040, C_22_H_32_O_7_Na); ^1^H NMR (600 MHz) and ^13^C NMR (125 MHz) data in CDCl_3_; see Table 1 and Table 2.

Penicilazaphilone J (**2**): yellow oil; [*α*]^25^_D_ −223.3 (*c* 0.1, MeOH); UV (MeOH) *λ*_max_ (log*ε*) 245 (3.03), 350 (3.75) nm; IR (KBr) *ν*_max_ 3445, 2930, 1744, 1659, 1586, 1378, 1230, 1061 cm^−1^; HRESIMS *m/z*: 839.4156 [2M + Na]^+^ (calcd. for 839.4188, C_44_H_64_O_14_Na); ^1^H NMR (600 MHz) and ^13^C NMR (125 MHz) data in CDCl_3_; see Table 1 and Table 2.

Epi-geumsanol D (**3**): yellow oil; [*α*]^25^_D_ −114.8 (*c* 0.1, MeOH); UV (MeOH) *λ*_max_ (log*ε*) 246 (2.93), 350 (3.53) nm; IR (KBr) *ν*_max_ 3433, 2930, 1641, 1584, 1380, 1232, 1062 cm^−1^; HRESIMS *m/z*: 417.1863 [M + Na]^+^ (calcd. for 417.1884, C_21_H_30_O_7_Na); ^1^H NMR (400 MHz) and ^13^C NMR (125 MHz) data in CDCl_3_; see Table 1 and Table 2.

Penidioxolane C (**4**): yellow oil; [*α*]^25^_D_ −113.0 (*c* 0.1, MeOH); UV (MeOH) *λ*_max_ (log*ε*) 244 (2.90), 351 (3.57) nm; IR (KBr) *ν*_max_ 3451, 2932, 1746, 1587, 1376, 1225, 1103 cm^−1^; HRESIMS *m/z*: 463.2683 [M + H]^+^ (calcd. for 463.2690, C_26_H_39_O_7_); ^1^H NMR (600 MHz) and ^13^C NMR (125 MHz) data in CDCl_3_; see Table 1 and Table 2.

Penidioxolane D (**5**): yellow oil; [*α*]^25^_D_ −170.3 (*c* 0.1, MeOH); UV (MeOH) *λ*_max_ (log*ε*) 245 (2.95), 349 (3.63) nm; IR (KBr) *ν*_max_ 3417, 2928, 1648, 1587, 1376, 1229, 1101 cm^−1^; HRESIMS *m/z*: 463.2688 [M + H]^+^ (calcd. for 463.2690, C_26_H_39_O_7_); ^1^H NMR (600 MHz) and ^13^C NMR (125 MHz) data in CDCl_3_; see Table 1 and Table 2.

Penicilazaphilone K (**6**): yellow oil; [*α*]^25^_D_ −161.1 (*c* 0.1, MeOH); UV (MeOH) *λ*_max_ (log*ε*) 245 (3.01), 349 (3.54) nm; IR (KBr) *ν*_max_ 3420, 2924, 1708, 1594, 1338, 1063 cm^−1^; HRESIMS *m/z*: 265.1073 [M + H]^+^ (calcd. for 265.1071, C_14_H_17_O_5_); ^1^H NMR (600 MHz) and ^13^C NMR (125 MHz) data in CDCl_3_; see Table 1 and Table 2.

Penicilazaphilone L (**7**): yellow solid; [*α*]^25^_D_ −81.5 (*c* 0.1, MeOH); UV (MeOH) *λ*_max_ (log*ε*) 246 (2.82), 350 (3.43) nm; IR (KBr) *ν*_max_ 3431, 2925, 1661, 1602, 1376, 1231, 1056 cm^−1^; HRESIMS *m/z*: 307.1180 [M + H]^+^ (calcd. for 307.1176, C_16_H_19_O_6_); ^1^H NMR (600 MHz) and ^13^C NMR (125 MHz) data in CDCl_3_; see Table 1 and Table 2.

Penicilazaphilone M (**8**): yellow oil; [*α*]^25^_D_ −215.0 (*c* 0.1, MeOH); UV (MeOH) *λ*_max_ (log*ε*) 213 (2.73), 323 (3.38) nm; IR (KBr) *ν*_max_ 3427, 2923, 1717, 1601, 1232, 1038 cm^−1^; HRESIMS *m/z*: 331.1141 [M + Na]^+^ (calcd. for 331.1152, C_16_H_20_O_6_Na); ^1^H NMR (600 MHz) and ^13^C NMR (125 MHz) data in CDCl_3_; see Table 1 and Table 2.

Penicilazaphilone N (**9**): yellow solid; [*α*]^25^_D_ +68.2 (*c* 0.1, MeOH); UV (MeOH) *λ*_max_ (log*ε*) 229 (3.24), 290 (3.28), 372 (3.20) nm; IR (KBr) *ν*_max_ 3460, 2925, 1773, 1612, 1250, 1090 cm^−1^; HRESIMS *m/z*: 351.0835 [M + Na]^+^ (calcd. for 351.0839, C_18_H_16_O_6_Na); ^1^H NMR (600 MHz) and ^13^C NMR (125 MHz) data in CDCl_3_; see Table 1 and Table 2.

### 3.4. NMR Calculation

Conformers of compounds were generated using the Confab [29] program ebbed in the Openbabel 3.1.1 software. All the conformers were further optimized with xtb at GFN2 level [30] and the conformers with population over 1% were subjected to geometry optimization using the Gaussian 16 package [31] at B3LYP/6-31G (d) level. The obtained conformers within an energy window of 3 kcal/mol were kept. Then, these conformers were refined by re-optimizations at B3LYP-D3BJ/6-311G (d,p) with IEFPCM solvent model in chloroform. Frequency analysis of all optimized conformations were also performed at the same level of theory to exclude the imaginary frequencies. NMR shielding tensors were calculated with the GIAO method at mPW1PW91/6-31 + G (d,p) level with IEFPCM solvent modeling in chloroform. The shielding constants were converted into chemical shifts by referencing to TMS at 0 ppm *(δ*cal = *σ*TMS − *σ*cal), where the *σ*TMS (the shielding constant of TMS) was calculated at the same level. Considering the almost same chemical shift of the azaphilone moiety, we only used the chemical shift of the side chain of **1**–**4** for the analysis to decrease the systematic error. For each candidate, the parameters a and b of the linear regression *δ*cal = a*δ*exp + b; the correlation coefficient, R2; the mean absolute error (MAE) defined as Σn |*δ*cal − *δ*exp|/n; the corrected mean absolute error, CMAE, defined as Σn |*δ*corr – *δ*exp|/n, where *δ*corr = (*δ*cal − b)/a, were calculated. DP4+ probability analysis was performed using the calculated NMR shielding tensors [23].

### 3.5. ECD Calculation

The conformers of compounds were generated using the Confab [29] program ebbed in the Openbabel 3.1.1 software, and further optimized with xtb at GFN2 level [30]. The conformers with population over 1% were subjected to geometry optimization using the Gaussian 16 package [31] at B3LYP/6-31G (d) level and proceeded to calculation of excitation energies, oscillator strength, and rotatory strength at B3LYP/TZVP level in the polarizable continuum model (PCM, methanol). The ECD spectra were Boltzmann weighted and generated using SpecDis 1.71 software [32].

### 3.6. Anti-Inflammatory Activity Test

All compounds were evaluated for their inhibitory effects on NO production in LPS-stimulated RAW264.7 macrophages (Stem Cell Bank of the Chinese Academy of Sciences, Shanghai, China) using the Griess assay [33]. The cells were cultured in DMEM medium (Thermo Fisher scientific, Waltham, MA, USA) in a humidified 5% CO_2_/95% air atmosphere at 37 °C. Each compound was diluted in half by concentration gradients (200 μM, 100 μM, 50 μM, 25 μM, 12.5 μM). Quercetin (Sigma Company, St. Louis, MO, USA) was used as positive controls and media with DMSO as negative control. The effects on cell viability of compounds was measured using MTT method.

### 3.7. Cytotoxic Detection

All compounds were assayed for their cytotoxic activities against five human tumor cell lines: human myeloid leukemia cells (K562), human liver cancer cells (BEL-7402), human gastric cancer cells (SGC-7901), human non-small cell lung cancer cells (A549), and human hela cervical cancer cells which were bought from the Cell Bank of Type Culture Collection of Shanghai Institute of Cell Biology, Chinese Academy of Sciences, using modified MTT methods [34]. Briefly, the abovementioned human tumor cell lines were cultured in RPMI-1640 medium with 10% FBS under a humidified atmosphere of 5% CO_2_ and 95% air at 37 °C, and 198 μL of cell suspension was plated in 96-well microtiter plates. An amount of 2 μL of the test solutions in DMSO was added to each well and further incubated for 36 h after being incubated for 24 h. An amount of 20 μL of MTT solution (5 mg/mL in RPMI-1640 medium) was added to each well and incubated for 4 h. Finally, 150 μL of medium containing MTT was gently replaced by DMSO and pipetted to dissolve any formazan crystals formed. Absorbance was tested on a Multiskan FC photometric microplate reader (Thermo Fisher Scientific) at 570 nm. Cisplatin was used as a positive control.

## 4. Conclusions

In summary, the investigation of *P*. *sclerotiorum* E23Y-1A derived from the marine sponge *Holoxea* sp. resulted in the isolation of nine new azaphilones (**1**–**9**), of which compounds **1**–**5** were identified as azaphilones possessing a branched C_7_ side chain fused at C-3. Compounds **6**–**9** were characteristic of the branded C_4_ side chain fused C-3, which are uncommonly found in nature. The bioassay of anti-inflammatory activity results revealed that compound **9** exhibited moderate inhibition of nitric oxide production in LPS-stimulated RAW264.7 cells, with an IC_50_ value of 22.63 ± 2.95 μM, and compounds **1**, **6** and **7** exhibited weak inhibition with IC_50_ values of 50.71 ± 8.41, 65.30 ± 7.21, and 31.84 ± 2.79 μM, respectively. The test of cytotoxic activities showed that penidioxolane C (**4**) exhibited moderate inhibition against K562, BEL-7402, SGC-7901, A549, and Hela cancer cells with IC_50_ values of 23.94 ± 0.11, 60.66 ± 0.13, 46.17 ± 0.17, 60.16 ± 0.26, and 59.30 ± 0.60 μM, respectively.

## Data Availability

The authors confirm that the data supporting the findings of this study are available within the article and its Appendix A.

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
