# Peer review of "New Azaphilones from the Marine-Derived Fungus Penicillium sclerotiorum E23Y-1A with Their Anti-Inflammatory and Antitumor Activities"

_marinedrugs, 2023, doi:10.3390/md21020075_

Round 1

Reviewer 1 Report

In the present manuscript the authors describe the structural elucidation and the biological evaluation of a series of new azaphilones isolated from a fungal strain of Penicillium species.

They made use of monodimensional and bidimensional NMR techniques, which together with Mass Spectrometry enabled the determination of the planar structure of the reported molecules which, by the way, are strictly related to other already known compounds.

In addition, they also proposed the stereochemistry of the above compounds; however, they used an inadequate approach to assign the relative, and consequently the absolute configuration of the stereocenters present in the flexible side chain of the compounds 1-5.

It’s well known that the determination of the stereochemistry of flexible acyclic carbon chain is not a simple task owing to the presence of multiple conformers in which, often, minor populations can make a large contribution to NOE intensity leading to misleading distance constrains.

Basically, for a robust stereochemistry determination in flexible moieties, NOE based methods can be safely used only in combination with QM calculations which help to determine the population ratio among conformers; in alternative other valid approach can be used as, for example, the J-based configuration analysis. In this regard, besides the plethora of papers dealing with this issue present in literature, I would mention a recent minireview appeared just on Marine Drugs (Mar.Drugs 2022, 20, 333).

Given that, in my opinion, the authors must provide a more confident approach to assign the stereochemistry of the side chains of compounds 1-5. As an alternative, I suggest living unassigned the configuration of the above mentioned stereocenters.

Reviewer 2 Report

The authors isolated new azaphilones from marine fungus, and they studied cellular anti-inflammatory and anticancer activities. my comments are as below.

1. Anti-inflammatory activity was measured by NO production in RAW264.7 cells. Results of NO production and MTT should be shown in figures with the dose effects of chemicals.

2. The cellular toxicity should be shown in Table.

Author Response

Please see the attached letter.

Reviewer 3 Report

This manuscript described 9 new azaphilones from a marine derived fungal culture with moderate NO inhibitory activities and cytotoxicities. The overall results are somehow interesting. However, there have some major concerns about the structure elucidations. Some methods and theory to determine the stereochemistry is wrong and must be corrected before consideration of acceptance.

The relative configurations of the pyranoquinone bicyclic core could be proposed by NOE correlations and 1H 1H coupling constants, however, the side chain should not use this approach due to the character of the free rotation of C-C bond. The side chain stereochemistry in the manuscript has a great possibility of wrong. It’s hard to believe different NOE correlations were observed between compound 2 and 3. It’s also un-precise to determine a structure solely based on a missing NOE correlation. In addition, in the structural elucidation part, the authors did not quote any references!

P4line 96: A large coupling constant is not enough to determine orientations of the two protons. Please add a reference.

The calculated ECD of 1 is not consistent with experimental CD curve in figure S1a, the curve of 0-250 nm are almost reverse.

Some NOE correlations do not make sense either, for compound 4, NOE between H-21/Me-18? From fig 3, H-21/Me-18 is too far away from each other.

Fig 4, ECD curves or CD curves? Calculated curve should be added for each compound.

Introduction should be expanded on the reported structures and activities of azaphilones.

The biosynthetic pathway of long and short side chain should be proposed.

It’s confusing that a part of the COSY, HMBC and NOE figures were in the SI.

Author Response

Please see the attached letter.

Round 2

Reviewer 1 Report

I read the Author replay; however, I keep on standing by my opinion; in the specific case the only use of 3JH-H is not adequate to discriminate among the conformers and that doesn’t enable the assignment of the relative configuration. Since 1999, with the introduction of Murata J-based configuration analysis, the combined use of 3JH-H and 2,3JC-H can enable the identification of the predominant rotamer out of the six possible staggered conformers from threo and erythro configurations, which makes possible to establish the relative configuration of flexible acyclic carbon chain.

Reviewer 2 Report

I still do not understand Fig 6., because graph and IC50 do not match. For example, IC50 of Comound 1 is about 1.7 microM from the graph. But the calculated value is about 30 microM. Also The toxicity table is difficult to understand. ND means over some microM?

Round 3

Reviewer 1 Report

The new data added by the authors (QM calculations analysed with DP4+ method) provided a more confident approach for the assignment of the relative stereochemistry of the described compounds. In my opinion the manuscript is now suitable for publication.